# Association between dental scaling and metabolic syndrome and lifestyle

TaeYeon Lee[1], Kyungdo Han[2], Kyoung-In Yun[3]*

1 Department of Conservative Dentistry, Yeouido St. Mary's Hospital, College of Medicine, The Catholic University of Korea, Seoul, Republic of Korea, 2 Department of Statistics and Actuarial Science, Soongsil University, Seoul, Republic of Korea, 3 Department of Oral and Maxillofacial Surgery, Yeouido St. Mary's Hospital, College of Medicine, The Catholic University of Korea, Seoul, Republic of Korea

* yun_ki@catholic.ac.kr

**Data Availability Statement:** Data cannot be shared publicly because of the legal restrictions, concerns for patient privacy, and third party owner ship of the data by the Korea National Health Insurance Service (NHIS). However, the

## Abstract

### Purpose

Periodontal disease is a risk factor for diabetes and metabolic syndrome, and non-surgical periodontal treatment has been shown to help maintain stable blood sugar in diabetic patients. Determining the level of preventive scaling in patients with metabolic syndrome will help manage the disease. The purpose of this study was to investigate the extent to which people with metabolic syndrome or bad lifestyle performed scaling and the association between preventive scaling and metabolic syndrome or lifestyle in a large population.

### Methods

This study was conducted on adults aged 20 years or older from January 2014 to December 2017 in the National Health Insurance System (NHIS) database. Among 558,067 people who underwent health checkups, 555,929 people were included. A total of 543,791 people were investigated for preventive scaling. Metabolic syndrome components were abdominal obesity, lower high density lipoprotein cholesterol (HDL)-C, high triglycerides, high blood pressure and hyperglycemia. Unhealthy lifestyle score was calculated by assigning 1 point each for current smokers, drinkers, and no performing regular exercise.

### Results

When multiple logistic regression analysis was performed after adjusting for age, sex, income, body mass index (BMI), smoking, drinking and regular exercise, the Odds ratios (OR) and 95% confidence intervals (CI) of the group with 5 metabolic syndrome components were 0.741 (0.710, 0.773) (p<0.0001). After adjustment for age, sex, income, BMI, smoking, drinking, regular exercise, diabetes, hypertension and dyslipidemia, the OR (95% CI) of the group with unhealthy lifestyle score = 3 was 0.612 (0.586, 0.640) (p<0.0001).

introduction and details of dataset are obtainable by accessing the Korean National Health Insurance Service homepage at https://nhiss.nhis.or.kr/. It can also be requested via nhiss@nhis.or.kr, +82-1577-1000, or from 32 Gungang-ro, Wonju-si, Gangwon-do 26464.

**Funding:** This research was supported by grant of the Institute of Clinical Medicine Research in the Yeouido St. Mary's Hospital, College of Medicine, The Catholic University of Korea(Grant Number: YSI22H007). The funder had no role in the study design, data collection and analysis, decision to publish, or preparation of the manuscript.

**Competing interests:** The authors have declared that no competing interests exist.

## Conclusions

The more metabolic syndrome components, and the higher unhealthy lifestyle score, the less scaling was performed.

## Introduction

Periodontal inflammation results from the interaction of the host immune system and a dysbiotic subgingival plaque biofilm and modified by lifestyle and environmental factors [1, 2]. Although plaque deposition is not a major risk factor for periodontitis, periodontal disease can be initiated by bacterial plaque [1, 3]. Microbiota in plaque biofilms release the inflammatory tissue breakdown products into the gingival crevice and inflammatory reactions in gingival sulcus can induce microbiota imbalance (dysbiosis) [4]. Dysbiosis can increase the inflammatory potential and ultimately cause the periodontitis in susceptible persons [4]. If the initial subgingival infection and tissue damage localized to the gingiva are not resolved, the epithelial barrier may be destroyed and clinical loss of periodontal attachment can occur [1].

If the bacterial plaque is not removed and remains on the tooth surface for a long time, it becomes calcified and becomes calculus [3]. The mineralized mass of gingival calculus is actually very porous and may play a secondary role in bacterial retention and a reservoir of various endotoxins [2, 5]. Because surface biofilm is always present, it is difficult to determine the causal relationship of subgingival calculus to the initiation and progression of periodontal disease [2]. Nevertheless, the presence of calculus is closely associated with inflammation of the subgingival pocket walls [2].

Periodontal disease is initiated by bacterial biofilm, but clinical manifestations vary depending on the individual's host inflammatory response and other predisposing factors such as lifestyle and systemic conditions. Various clinical studies have reported increased susceptibility and severity of periodontitis and more severe periodontal disease progression in smokers compared to non-smokers, suggesting that smoking-induced alterations in the host's inflammatory response and microbiological changes may be involved [6, 7]. The negative effects of smoking on periodontitis can be further exacerbated by excessive drinking [7].

Periodontal pathogens do not remain only in tissues around the teeth, but can move through the bloodstream and cause low-grade inflammation throughout the body. Various previous studies have estimated that periodontal disease acts as a risk factor for cardiovascular disease, diabetes, and metabolic syndrome [8–11]. Patients with periodontitis are known to have more severe endothelial dysfunction, increased arterial calcification scores, and increased risk of myocardial infarction compared to patients without periodontitis [11]. Glycated hemoglobin increased over time in people with periodontitis [9]. The presence of periodontal pockets was associated with increased metabolic syndrome components [9]. However, the association between metabolic syndrome and periodontitis could be bi-directional. Diabetes is a high-level risk factor for gingivitis and periodontitis, and glycaemia and periodontal disease risk show a dose-response relationship [12]. Periodontitis seems to occur in conjunction with a number of diabetes-related complications and is even considered a complication of diabetes [13]. It has been reported that people with metabolic syndrome are twice as likely to develop periodontitis than those without it [9].

As discussed above, many studies report an association between systemic diseases and periodontitis, but the causal relationship is not clear. This potential association between periodontitis and metabolic syndrome may be related to common risk factors such as smoking and diet. Smoking increases the risk of metabolic syndrome, regardless of the age of onset or duration

of smoking [14, 15]. Compared to non-smokers, smokers have more unhealthy lifestyle habits such as alcohol consumption and physical inactivity, which are known to increase the risk of metabolic disease [15]. A healthy lifestyle may prevent or delay the onset of metabolic syndrome in people who are susceptible to it.

Periodontal disease causes local and systemic inflammatory reactions, so proper prevention and treatment are very important. Treatment of periodontal disease includes non-surgical treatment and surgical treatment. Scaling is the gold standard of non-surgical periodontal treatment and a preventive treatment. Scaling alone can reduce bleeding and periodontal pockets in teeth with severe periodontitis [12]. It is known that the treatment of periodontal disease can improve not only the local inflammatory condition around the tooth but also the systemic inflammatory condition. After treatment for periodontal disease, systemic markers related to cardiovascular disease were reported to decrease [12]. Non-surgical periodontal treatment has been shown to reduce HbA1C levels in diabetic patients and help maintain stable blood sugar [16]. After regular scaling, surgical site infection in knee arthroplasty patients was reported to be reduced [17].

Dental scaling not only serves as a treatment for local inflammation around the teeth, but also plays an important role in the treatment of various systemic diseases and reduction of complications. However, most of the studies so far have observed changes in clinical indicators or test values related to the disease after scaling on patients with systemic diseases. There has been no study on the relationship between the scaling performed for preventive purposes and metabolic syndrome or lifestyle in a large population.

The purpose of this study was to investigate the extent to which people with metabolic syndrome or bad lifestyle performed scaling, a basic periodontal disease management program and the association between preventive scaling and metabolic syndrome or lifestyle in a large population.

## Methods

### Study population and design

In this study, the National Health Insurance Service-National Sample Cohort (NHIS-NSC) database produced by the National Health Insurance System (NHIS) was used. NHIS is the single mandatory health insurance program that covers approximately 97% of the Korean population. NHIS-NSC was launched in 2000 by integrating 375 insurance societies, and provides longitudinal data on 97% of the Korean population in connection with the national death registry and national health examination program [18, 19]. The remaining 3% of the low-income population is covered by health assistance programs and their data have been integrated into the NHIS database. Therefore, since 2006, the NHIS database has included information on comorbidities, drug prescriptions, treatment claims, and demographic characteristics of virtually the entire population of Korea. The NHIS database is based on the ICD-10-CM (International Classification of Disease, Tenth Revision, Clinical Modification) codes. The National Health Screening program, launched in 2009 and administered by the government, includes a medical interview and blood pressure test, chest X-ray, blood test, urine test, and dental exam. Health checkups are managed by the government and are practiced by almost all Koreans. NHIS-NSC data can only be analyzed by the analysis office within the National Health Insurance Service for studies that have passed the study protocol, and the original data are deleted after a certain period of time. (https://nhiss.nhis.or.kr/bd/ab/bdaba000eng.do;jsessionid=vlpHYbY2GAeotWKtU4XI9S8JvdHRUXclZxYlCu25G8htHNZaxz4zeBP3Q1VDIaSs.primrose22_servlet_engine10).

From July 2013, adults 20 years of age and older are covered by the NHIS-NSC for preventive scaling once a year. Scaling for preventive purposes is performed to prevent periodontal disease or to prevent deterioration of it, and means to be performed when additional periodontal treatment is not required after scaling. Scaling for therapeutic purposes is defined as being performed before curettage or periodontal surgery. Scaling for preventive purposes is managed as a separate prescription code to distinguish it from scaling for therapeutic purposes. This study was conducted on adults aged 20 years or older from January 2014 to December 2017 in the NHIS-NSC database. Those under 20 years of age or missing data were excluded. From January 2014 to December 2017, among 558,067 people who underwent health checkups, 555,929 people aged 20 years or older were included. A total of 543,791 people, excluding double missing persons, were investigated for preventive scaling considering various confounders.

## Variables

Age, gender, and family income level were used as sociodemographic variables. Household income was corrected for the number of family members and divided into quintiles: < 20%, 20%~39%, 40%~59%, 60~79% and > 80% of the total equivalized income in the survey. We consider < 20% as low-income group. From the health behaviors survey data, we utilized smoking, drinking, and regular exercise. The drinking status was classified to never, mild (<30g alcohol per day) and heavy (≥30g alcohol per day). The smoking status was categorized into three groups: non-smoker, ex-smoker (those who had smoked in the past but had ceased smoking) or current smoker. The regular exercise was defined as moderate-intensity physical activity for at least 30 minutes at least 5 days a week or high-intensity exercise for at least 20 minutes at least 3 days a week.

Body weight and height were measured with the participants in light in door clothing without shoes. Waist circumference (WC) was measured at the narrowest point between the lower border of the rib cage and the iliac crest. Body mass index (BMI) was calculated by the following formula: weight/height$^2$ (kg/m$^2$). Systolic and diastolic blood pressure (BP) were measured on the right arm two times with five-minute interval, and the average value were used for the analysis. A blood sample was collected from antecubital vein of each subject after fasting for more than eight hours. Serum glucose, total cholesterol, triglyceride, high density lipoprotein (HDL) cholesterol, and low-density lipoprotein (LDL) cholesterol were measured. Hypertension was defined as ICD-10 codes I10 to I13 or I15 and treatment with antihypertensive medications, systolic BP ≥ 140mmHg, or diastolic BP ≥ 90mmHg). Type 2 diabetes was defined as ICD-10 codes E11 to E14 and anti-diabetic drugs, or fasting glucose level ≥ 126mg/dL) and hyperlipidemia was defined as ICD-10 code E78 with lipid-lowering agents, or serum total cholesterol ≥ 240mg/dL.

Metabolic syndrome was defined based on the modified criteria of the National Cholesterol Education Program Adult Treatment Panel III, while abdominal obesity was based on the Asian-specific WC cutoff [20]. Individuals were diagnosed with Metabolic syndrome if they had three or more of the following: (1) a WC of ≥ 90 cm for men and ≥ 85 cm for women [21]; (2) a serum triglyceride level of ≥ 150 mg/dL or were treated with lipid-lowering medication; (3) a serum HDL-C level of < 40 mg/dL for men or < 50 mg/dL for women; (4) a BP of ≥ 130/85 mmHg or were treated with antihypertensive medication; and (5) a fasting blood glucose level of ≥ 100 mg/dL or were treated with antidiabetic medication.

Based on the results of preliminary analyses, unhealthy lifestyle was defined as smoking, alcohol consumption, and non-regular exercise [22]. Unhealthy lifestyle score was calculated by assigning 1 point each for current smokers, drinkers, and no performing regular exercise.

The study population was categorized into four groups according unhealthy lifestyle score from patients with score 0 who did not have any unhealthy lifestyle behaviors to patients with score 3 who met all three unhealthy lifestyle behaviors.

Scaling for preventive purposes is managed with a separate prescription code from scaling for treatment purposes. Because the outcome variable for this study was preventive dental scaling, prescription code for therapeutic scaling, regardless of referral, was excluded. The outcome variable, preventive dental scaling, was defined as whether or not the person had a preventive scaling in the same year as the year of the health checkup (Binary variable).

### Statistical analyses

Baseline characteristics were analyzed using descriptive statistics. Categorical variables were described as frequencies (%). Continuous variables were described as means ± standard deviation (SD). An independent t-test for continuous variables or a chi-square test of categorical variables were used to analyze the relationship with scaling.

Multiple logistic regression analysis was used to assess the association between metabolic syndrome, lifestyle and scaling. Odds ratios (OR) and 95% confidence intervals (CI) were estimated after adjustment for potential confounders. The variables in the final model were age, sex, BMI, income level, smoking, alcohol consumption, regular exercise, diabetes, hypertension, and dyslipidemia (variables in Table 2). The variables in the final model were grouped into groups (demographics, body composition, socio-economics, lifestyle, comorbidities). The variables were added to the model progressively from non-adjusted to fully adjusted.

Three types of model were used to multiple logistic regression analysis according to metabolic syndrome. Model 1 was not adjusted. Model 2 was adjusted for demographic variables (age and sex). Model 3 was adjusted for the variables in model 2 plus body composition variable (BMI) and socio-economic variables (income level) and lifestyle variables (smoking, alcohol consumption and regular exercise). Four types of model were used to multiple logistic regression analysis according to lifestyle. Model 1 was not adjusted. Model 2 was adjusted for age and sex (demographic variables). Model 3 was adjusted for the variables in model 2 plus income level (socio-economic variables), and BMI (body composition variable). Model 4 was adjusted for the variables in model 3 plus comorbidities such as diabetes, hypertension and dyslipidemia (Fig 1). We used SAS version 9.2 (SAS institute Inc., Cary, NC, USA) for statistical analysis. Two-sided *P*- values of < 0.05 were considered statistically significant,

## Results

The general characteristics of the population categorized by with and without scaling are shown in Table 1. Among 543,791 people, 105,966 people were in the group that performed scaling for preventive purposes. The mean age of the scaling group was lower than that of the non-scaling group (50.29±14.91 vs. 46.34±13.11, p<0.0001). The prevalence of diabetes, hypertension, and dyslipidemia was lower in the scaling group than in the non-scaling group (p<0.0001). Systolic and diastolic blood pressure and fasting blood glucose levels were lower in the scaling group (p<0.0001). There was no difference in weight between the two groups, but BMI and waist circumference were lower in the scaling group (p<0.0001). The regular exercise execution rate was higher in the scaling group than in the non-scaling group.

Table 2 shows the OR (95% CI) results of multivariate analysis for scaling trials. The lower the income and the older the age, the less scaling (p<0.0001). Males were doing less scaling than females (p<0.0001). Groups with BMI < 18.5 or BMI ≥ 23 received less preventive scaling, but in the case of BMI ≥ 23, as the BMI level increased, they received less preventive

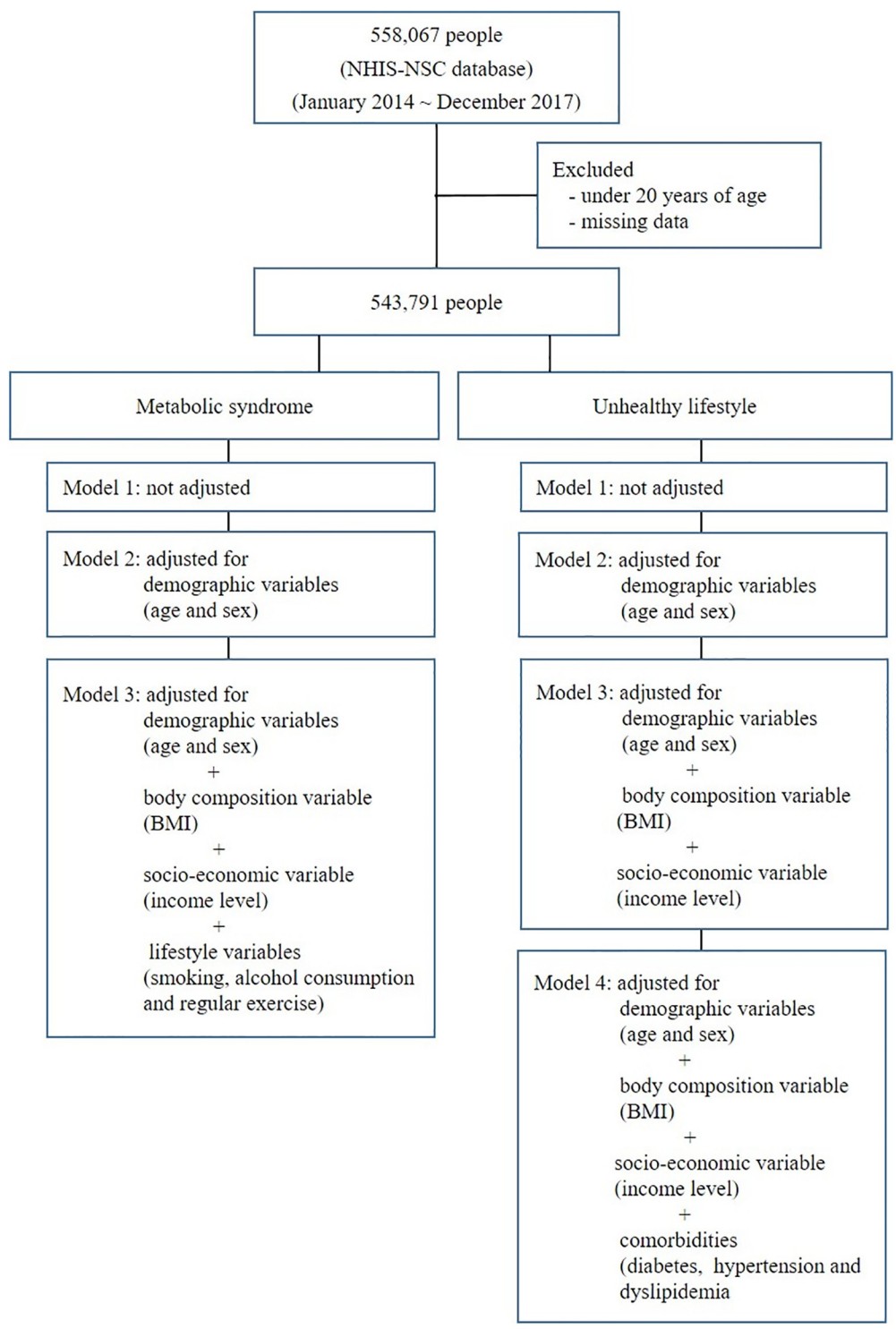

**Fig 1. Study population and design.**

**Table 1. The clinical characteristics of study participants.**

| | Preventive scaling | | |
| --- | --- | --- | --- |
| | **No** | **Yes** | **p-value** |
| | **(n = 437,825)** | **(n = 105,966)** | |
| Age groups | | | < 0.0001 |
| < 40 | 104,807 (23.94) | 32,009 (30.21) | |
| 40–64 | 254,166 (58.05) | 64,829 (61.18) | |
| ≥ 65 | 78,852 (18.01) | 9,128 (8.61) | |
| Age, years | 50.29±14.91 | 46.34±13.11 | < 0.0001 |
| Sex | | | < 0.0001 |
| Male | 218,627 (49.93) | 50,933 (48.07) | |
| Female | 219,198 (50.07) | 55,033 (51.93) | |
| Income, Lower 20% | 75,137 (17.16) | 15,766 (14.88) | < 0.0001 |
| Body mass index (BMI) Level | | | < 0.0001 |
| < 18.5 | 16,597 (3.79) | 4,272 (4.03) | |
| < 23‘ | 162,913 (37.21) | 42,992 (40.57) | |
| < 25 | 103,212 (23.57) | 24,860 (23.46) | |
| < 30 | 132,631 (30.29) | 29,530 (27.87) | |
| ≥ 30 | 22,472 (5.13) | 4,312 (4.07) | |
| Smoking | | | < 0.0001 |
| Non | 271,372 (61.98) | 67,852 (64.03) | |
| Ex | 67,259 (15.36) | 17,808 (16.81) | |
| Current | 99,194 (22.66) | 20,306 (19.16) | |
| Drinking | | | < 0.0001 |
| Non | 231,298 (52.83) | 52,953 (49.97) | |
| Mild | 172,676 (39.44) | 45,839 (43.26) | |
| Heavy | 33,851 (7.73) | 7,174 (6.77) | |
| Regular exercise | 83,472 (19.07) | 22,762 (21.48) | < 0.0001 |
| Diabetes mellitus | 51,868 (11.85) | 8,269 (7.8) | < 0.0001 |
| Hypertension | 128,111 (29.26) | 22,076 (20.83) | < 0.0001 |
| Dyslipidemia | 113,820 (26) | 24,358 (22.99) | < 0.0001 |
| Systolic blood pressure, mmHg | 122.65±15 | 119.87±14.07 | < 0.0001 |
| Diastolic blood pressure, mmHg | 76.12±10.03 | 74.86±9.77 | < 0.0001 |
| Fasting glucose, mg/dL | 100.37±25.79 | 97.29±20.83 | < 0.0001 |
| Total Cholesterol, mg/dL | 194.63±37.94 | 194.72±36.92 | 0.4936 |
| HDL -C, mg/dL | 55.83±15.38 | 57.07±16.07 | < 0.0001 |
| LDL -C, mg/dL | 113.61±35.66 | 113.91±34.41 | 0.0115 |
| Height, cm | 163.53±94 | 164.57±8.89 | < 0.0001 |
| Weight, kg | 64.34±12.46 | 64.37±12.35 | 0.4566 |
| BMI, kg/m$^2$ | 23.95±3.48 | 23.64±3.35 | < 0.0001 |
| Waist Circumference, cm | 80.87±9.68 | 79.62±9.66 | < 0.0001 |

Data are presented as frequencies (%) in categorical variables and means ± standard deviation (SD) in continuous variables.

P-values were obtained by an independent t-test for continuous variables or a chi-square test of categorical variables. P< 0.05 is statistically significant.

HDL, high density lipoprotein cholesterol; LDL, low-density lipoprotein cholesterol

scaling (p<0.0001). Ex-smokers were found to be scaling more than non-smokers, but current smokers were found to be scaling less (p<0.0001). Mild drinkers performed more scaling than non-drinkers, but heavy drinkers did less (p<0.0001). The group that exercised regularly performed more preventive scaling than the group that did not (p<0.0001). People with diabetes,

**Table 2. Association of clinical characteristics with scaling in univariate analyses.**

| | Event N (%) | OR (95% C.I) | p-value |
|---|---|---|---|
| Age groups | | | < 0.0001 |
| < 40 | 32,009 (23.4) | 1 (Ref.) | |
| 40~64 | 64,829 (20.32) | 0.835 (0.823, 0.848) | |
| ≥ 65 | 9,128 (10.38) | 0.379 (0.370, 0.389) | |
| Sex | | | < 0.0001 |
| Male | 50,933 (18.89) | 0.928 (0.916, 0.940) | |
| Female | 55,033 (20.07) | 1 (Ref.) | |
| Income | | | < 0.0001 |
| Quintile 1 | 15,766 (17.34) | 1 (Ref.) | |
| Quintile 2 | 16,497 (18.4) | 1.075 (1.049, 1.101) | |
| Quintile 3 | 19,569 (18.77) | 1.101 (1.076, 1.127) | |
| Quintile 4 | 23,827 (19.72) | 1.170 (1.145, 1.197) | |
| Quintile 5 | 30,307 (21.94) | 1.339 (1.311, 1.368) | |
| BMI Level | | | < 0.0001 |
| < 18.5 | 4,272 (20.47) | 0.975 (0.942, 1.010) | |
| < 23 | 42,992 (20.88) | 1 (Ref.) | |
| < 25 | 24,860 (19.41) | 0.913 (0.897, 0.929) | |
| < 30 | 29,530 (18.21) | 0.844 (0.830, 0.858) | |
| ≥ 30 | 4,312 (16.1) | 0.727 (0.703, 0.753) | |
| Smoking | | | < 0.0001 |
| Non | 67,852 (20) | 1 (Ref.) | |
| Ex | 17,808 (20.93) | 1.059 (1.039, 1.079) | |
| Current | 20,306 (16.99) | 0.819 (0.805, 0.833) | |
| Drinking | | | < 0.0001 |
| Non | 52,953 (18.63) | 1 (Ref.) | |
| Mild | 45,839 (20.98) | 1.160 (1.143, 1.176) | |
| Heavy | 7,174 (17.49) | 0.926 (0.901, 0.951) | |
| Regular exercise | | | < 0.0001 |
| No | 83,204 (19.02) | 1 (Ref.) | |
| Yes | 22,762 (21.43) | 1.161 (1.142, 1.181) | |
| Diabetes | | | < 0.0001 |
| No | 97,697 (20.2) | 1 (Ref.) | |
| Yes | 8,269 (13.75) | 0.630 (0.615, 0.645) | |
| Hypertension | | | < 0.0001 |
| No | 83,890 (21.31) | 1 (Ref.) | |
| Yes | 22,076 (14.7) | 0.636 (0.626, 0.647) | |
| Dyslipidemia | | | < 0.0001 |
| No | 81,608 (20.12) | 1 (Ref.) | |
| Yes | 24,358 (17.63) | 0.850 (0.836, 0.863) | |

CI, confidence interval; OR, odds ratio.

P< 0.05 is statistically significant.

hypertension, and dyslipidemia were doing less scaling than those without these conditions (p<0.0001).

Fig 2 shows the OR (95% CI) of multiple logistic regression analysis stratified by metabolic syndrome and classified by scaling presence. Even after adjusting for various variables, the

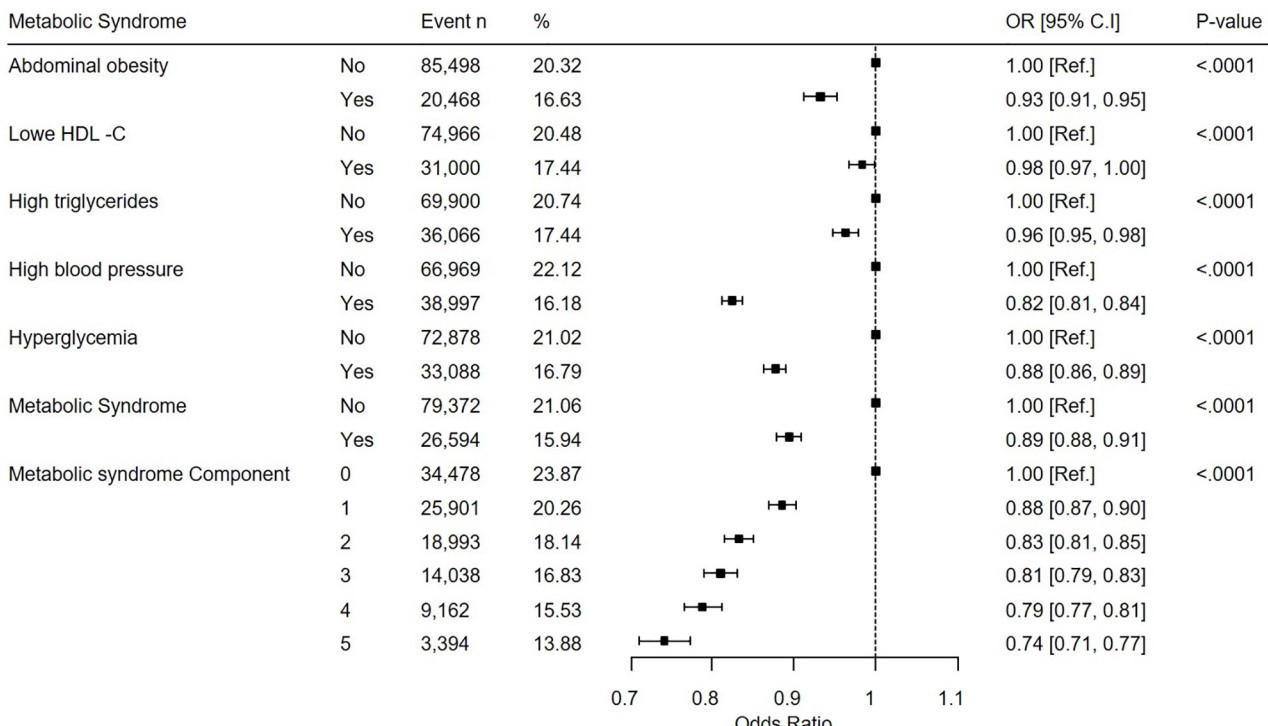

**Fig 2. Multiple logistic regression analysis according to metabolic syndrome.** P< 0.05 is statistically significant.

group with obesity, hypertension, diabetes, dyslipidemia, and metabolic syndrome showed less scaling, and the more metabolic syndrome components, the less scaling. As a result of analysis after adjustment for age, sex, income, BMI, smoking, drinking and regular exercise, the OR (95% CI) of the group with 5 metabolic syndrome components was 0.74 (0.71, 0.77) (p<0.0001).

This forest plot is the result of analysis by adjusting for age, gender, income, BMI, smoking, drinking, and regular exercise. This forest plot was created based on the results in Appendix Table 1 in S1 Appendix.

Fig 3 shows the OR (95% CI) of multiple logistic regression analysis stratified by lifestyle and classified by the presence or absence of scaling. Even after adjusting the variables, the higher the unhealthy lifestyle score, the lower the scaling. As a result of analysis after adjustment for age, sex, income, BMI, smoking, drinking, regular exercise, diabetes, hypertension and dyslipidemia, the OR (95% CI) of the group with unhealthy lifestyle score = 3 was 0.61 (0.59, 0.64) (p<0.0001).

This forest plot is the result of analysis by adjusting for age, sex, income, BMI, smoking, drinking, regular exercise, diabetes, hypertension and dyslipidemia. This forest plot was created based on the results in Appendix Table 2 in S1 Appendix.

## Discussion

As a result of adjusting and analyzing potential confounding variables, this study showed that the more metabolic syndrome components, and the higher the unhealthy lifestyle score, the less scaling was performed.

| Lifestyle | | Event n | % | | OR [95% C.I] | P-value |
|---|---|---|---|---|---|---|
| Smoking | Non | 67,852 | 20.00 | | 1.00 [Ref.] | <0.0001 |
| | Ex | 17,808 | 20.93 | | 1.10 [1.07, 1.12] | |
| | Current | 20,306 | 16.99 | | 0.78 [0.77, 0.80] | |
| Drinking | Non | 52,953 | 18.63 | | 1.00 [Ref.] | <0.0001 |
| | Mild | 45,839 | 20.98 | | 1.04 [1.03, 1.06] | |
| | Heavy | 7,174 | 17.49 | | 0.93 [0.90, 0.96] | |
| Regular exercise | No | 83,204 | 19.02 | | 1.00 [Ref.] | <0.0001 |
| | Yes | 22,762 | 21.43 | | 1.16 [1.14, 1.18] | |
| Unhealthy Lifestyle Score | 0 | 18,233 | 22.49 | | 1.00 [Ref.] | <0.0001 |
| | 1 | 67,765 | 19.60 | | 0.83 [0.81, 0.84] | |
| | 2 | 16,985 | 17.24 | | 0.67 [0.64, 0.69] | |
| | 3 | 2,983 | 16.19 | | 0.61 [0.59, 0.64] | |

Odds Ratio (0.5 0.7 0.9 1.1 1.3)

**Fig 3. Multiple logistic regression analysis according to lifestyle.** P< 0.05 is statistically significant.

Periodontitis is a chronic multifactorial inflammatory disease associated with dysbiotic plaque biofilms and characterized by the progressive destruction of the tooth-supporting apparatus [23, 24]. It's progression is related with individual's immunity and lifestyle, especially smoking and poor oral hygiene [25]. If the inflammatory response is not controlled, extensive periodontal destruction is occurred. The periodontal pocket epithelium is involved in systemic spread of local inflammation because of its direct contact with subgingival biofilm [25]. Pro-inflammatory cytokines and enzymes from periodontitis may migrate into the circulation and cause distant systemic inflammation although systemic inflammation can also contribute to periodontitis [25, 26]. Patients with periodontitis have higher values of white blood cells, interleukin (IL)-1, IL-6, tumor necrosis factor (TNF)-α and C-reactive protein (CRP) [25, 26]. Long-term retention of these cytokines may alter the systemic inflammatory state. In this way, periodontitis may exacerbate pre-existing systemic diseases such as diabetes and cardiovascular disease [25].

According to meta-analyses of the relationship between periodontitis and metabolic syndrome, the odds ratio between metabolic syndrome and periodontitis shows a slight difference between countries, but shows that periodontitis and metabolic syndrome are related [27, 28]. In addition, it was found that the relationship between periodontitis and metabolic syndrome increased as the severity of periodontitis and the number of metabolic syndrome factors increased [29]. Patients with metabolic syndrome had more severe periodontal disease than those without, and the pattern of oral bacteria was different [30, 31]. In the case of metabolic syndrome accompanied by periodontitis, more major bacteria of periodontal disease were detected than in the group with only metabolic syndrome, and the difference in treatment effect was significant [31]. Changes in oral flora were thought to increase the risk of metabolic syndrome due to periodontal disease [31].

Although the exact mechanism by which local inflammation caused by periodontal pathogens and cytokines induce systemic inflammation is unknown, it is thought that removing the cause of local inflammation can help treat diabetes and metabolic syndrome. Dental scaling is not only non-surgical periodontal therapy to control local inflammation around the teeth by mechanically removing bacterial plaque and calculus attached to the teeth, but also prophylactic treatment to prevent the occurrence and progression of local inflammation around the teeth. Removing the bacterial biofilm through dental scaling can help to stop the progression of inflammation in its earlier stages. Non-surgical periodontal therapy was associated with a progressive decrease in CRP in healthy individuals [32]. It has been reported that dental scaling affects the control of diabetes and hyperlipidemia. After scaling in patients with type 2 diabetes, high-sensitivity C-reactive protein (hs-CRP), IL-1ß, IL-6 concentrations and HbA1c decreased [33–35]. In patients with hyperlipidemia, pro-inflammatory cytokine levels and serum lipid levels decreased after periodontal disease treatment [36, 37]. Recent study reported that moderate-certainly evidence that periodontal therapy is helpful in reduction of HbA1c in diabetic patients [38].

As seen above, there are many previous studies showing that systemic diseases such as diabetes, cardiovascular disease, and metabolic syndrome improve disease-related values after scaling, so it is important to actively manage periodontal disease from the diagnosis stage of these diseases. The present study investigated the extent to which patients with metabolic syndrome performed scaling, a basic periodontal disease management program. According to this study, the group with diabetes, dyslipidemia, hypertension, obesity, and metabolic syndrome received less scaling than the group without these conditions. In addition, the more metabolic syndrome factors, the less preventive scaling was received. The results of this study indicate that the group with poorly controlled systemic diseases neglected oral hygiene management more than the well-controlled group. The results of this study cannot accurately determine the cause. However, judging from the results of this study, it is necessary to check whether the accessibility of dental treatment for patients with metabolic syndrome, including diabetes, has decreased, and whether there is a lack of education on the importance of oral hygiene management and regular scaling to prevent periodontal disease. Dental clinicians may ask physicians to recommend that patients with metabolic syndrome visit the dental clinics.

The present study also investigated the association between bad lifestyle and dental scaling. Lifestyle factors such as smoking and physical exercise affect the development and exacerbation of periodontitis as well as metabolic syndrome [3, 39]. It has been reported that people with a healthy lifestyle have a lower incidence of metabolic syndrome and periodontitis [39, 40]. According to this study, it was found that the higher the unhealthy lifestyle score, the less scaling was received.

Smoking is a major risk factor for periodontal disease [6]. Smoking may affect periodontal destruction through the microcirculatory, inflammatory and immune systems [6]. Smoking may lead to change of composition of the subgingival biofilm, formation of specific periodontal pathogen colonization and compromising the immune response [6, 41]. Many previous studies have shown that smokers have more periodontal tissue destruction than non-smokers and do not respond well to nonsurgical periodontal treatment [42]. However, scaling alone presents improving periodontal clinical parameters and reducing the levels of periodontal pathogens in smokers [6]. It is known that when smoking cessation periodontal pathogens change to the same pattern as non-smokers, and the effect of periodontal treatment is improved [6, 43]. In this study, it was found that current smokers received less preventive scaling. However, former smokers received more prophylactic scaling than non-smokers or current smokers. The cause of quitting smoking of former smokers is not known, but it is

presumed that attention to systemic and oral health may have contributed to their decision to stop smoking.

Alcohol consumption is also known to be associated with the development of periodontitis. Drinkers had a higher incidence of periodontitis than non-drinkers [44]. The risk of periodontitis increased as the amount of alcohol consumed or the frequency of drinking increased [44–46]. However, some authors reported that periodontal parameters of moderate drinkers were not different from non-drinkers [47, 48]. This result was thought to be related to the heterogeneity of assessment methods or same mechanism, down regulation of pro-inflammatory cytokines as the beneficial effects of light drinking on cardiovascular disease and type 2 diabetes [47]. In this study, it was found that mild drinkers received more preventive scaling than non-drinkers, but heavy drinkers received less preventive scaling than non-drinkers or mild drinkers. It seems that clinicians need to help heavy drinkers reduce the amount and frequency of alcohol consumption, not only for their systemic health, but also for their oral health. It also seems that dental clinicians need to find way to make it easier for heavy drinkers to participate in oral healthcare program including preventive scaling.

This study showed that the more metabolic syndrome components, and the higher unhealthy lifestyle score, the less scaling was performed. The results of this study suggest that clinicians need to find ways to make periodontal disease prevention and management programs more accessible to people with metabolic syndrome or unhealthy lifestyles.

## Limitations and strengths

This study has several limitations. First of all, it is difficult to generalize the results of this study because the survey data from only one Asian country. Second, because this study targeted people who had undergone a health examination registered in the NHIS database, those who had not undergone a health examination were not included. Third, the analysis period was short, about 4 years. Fourth, it is difficult to determine a causal relationship. Forth, because this study focused on preventive care, data from people who received only therapeutic scaling without preventive scaling were not included.

Nevertheless, this study has the following strengths. First, there is almost no bias based on culture and race because this study was conducted on a relatively ethnically homogenous population. Second, population-based data were used, in which more than 97% of the total population of Korea participated. Third, because this study used the results of a targeting a large population, it can provide epidemiological evidence that the more metabolic syndrome components, and the higher the unhealthy lifestyle score, the less scaling was performed.

## Conclusions

As a result of analysis after adjusting for potential confounding variables, it was found that the more metabolic syndrome components, and the higher unhealthy lifestyle score, the less scaling was performed. These findings suggest that plans are needed to increase access to preventive scaling for people with metabolic syndrome or unhealthy lifestyles.

## Supporting information

**S1 Appendix.**
(DOCX)

**S1 Checklist. STROBE statement—Checklist of items that should be included in reports of observational studies.**
(DOCX)

## Author Contributions

**Conceptualization:** TaeYeon Lee, Kyungdo Han, Kyoung-In Yun.

**Data curation:** Kyungdo Han.

**Formal analysis:** TaeYeon Lee, Kyungdo Han, Kyoung-In Yun.

**Funding acquisition:** Kyoung-In Yun.

**Investigation:** TaeYeon Lee, Kyungdo Han, Kyoung-In Yun.

**Methodology:** Kyungdo Han.

**Project administration:** TaeYeon Lee.

**Supervision:** TaeYeon Lee, Kyungdo Han, Kyoung-In Yun.

**Validation:** Kyoung-In Yun.

**Writing – original draft:** TaeYeon Lee.

**Writing – review & editing:** Kyoung-In Yun.

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
