## [Decision Letter · Decision Letter 0]

20 Jun 2023

PONE-D-23-15887Association between dental scaling and metabolic syndrome and lifestyle: A national cohort studyPLOS ONE

Dear Dr. Yun,

Thank you for submitting your manuscript to PLOS ONE. After careful consideration, we feel that it has merit but does not fully meet PLOS ONE’s publication criteria as it currently stands. Therefore, we invite you to submit a revised version of the manuscript that addresses the points raised during the review process. 

Dear authors,

Three experts in the field have reviewed your manuscript and raised serious concerns regarding the conceptual model underlying your study. Particular attention should be given to the presentation of the results, as all referees indicated issues related to lack of clarity and the exceeding number of tables. Please consider including a figure, as suggested by one of the referees. 

I look forward to receiving your revised manuscript. 

We look forward to receiving your revised manuscript.

Kind regards,

Gustavo G. Nascimento, PhD

Academic Editor

PLOS ONE

Journal Requirements:

Reviewers' comments:

Reviewer's Responses to Questions

**Comments to the Author**

1. Is the manuscript technically sound, and do the data support the conclusions?

Reviewer #1: No

Reviewer #2: Partly

Reviewer #3: Yes

2. Has the statistical analysis been performed appropriately and rigorously? 

Reviewer #1: I Don't Know

Reviewer #2: No

Reviewer #3: Yes

3. Have the authors made all data underlying the findings in their manuscript fully available?

Reviewer #1: No

Reviewer #2: No

Reviewer #3: No

4. Is the manuscript presented in an intelligible fashion and written in standard English?

Reviewer #1: No

Reviewer #2: Yes

Reviewer #3: No

5. Review Comments to the Author

Reviewer #1: The paper has the potential to enrich the literature by promoting a discussion about whether preventive scaling would indeed have any impact on indicators of metabolic syndrome in a population-based study.

I recommend that the authors reconsider the theoretical concept underlying the analyses and reorganize the presentation of the paper. In an epidemiological study like this, clarifying the outcomes and exposures according to your aims is crucial.

In a deep reflection, I suggest revisiting some assumptions still unclear in the literature. There is no consensus on the role of periodontitis in chronic disease causation. Studies show that, most likely, both share the same risk factors, such as an unhealthy diet. In this sense, preventive scaling would have little impact on periodontitis and even less on other metabolic risks.

Reviewer #2: The study uses a national database to report the associations of metabolic syndrome and unhealthy lifestyle (as exposures) and preventive scaling (as the outcome). The topic is clinically relevant as there is a need to identify the utilization rates of preventive dental care in such individuals at risk of periodontal disease. However, there are major methodological issues that need to be addressed before it can be considered for publication. Below are a few points for the consideration of the authors:

1. The statistical plan addresses the main objectives of the study using too many multivariable models and is rather confusing to follow. For instance, it is difficult to understand what Table 2 adds in the overall analysis, besides identifying the significant covariates associated with the outcome. If that is the case, the univariate analysis should suffice. As such, the multivariable analysis in Table 2 only adds to the ambiguity, for example, dyslipidemic individuals are shown to be at higher odds of preventive scaling which is in contrast to what has been reported in Table 3, where individuals with high triglycerides and low HDL-C levels are at lower odds of preventive scaling.

2. In Table 3, metabolic syndrome is categorized as a binary variable (yes/no), ordinal variable (with its five components together), and also as individual components tested separately with the outcome. The rationale of having so many definitions of metabolic syndrome to test this association is unclear. Also, the rationale behind having a stepwise inclusion of confounders as Models 1,2,3 needs to be clearly mentioned if it adds anything to the overall results, in contrast to reporting only model 3 which includes all the potential confounders.

3. Similarly, Table 4 also reports multiple models as well as definitions of exposure to test the association which either needs to be justified or needs to be trimmed down substantially. More so, some of the models are very perplexing. For example, Model 3 in Table 4 adjusts for smoking, drinking and regular exercise while testing the association of the same exposures with the outcome. The authors would need to re-organize the tables, to highlight main findings they believe helps to test their study objective.

4. Minor points:

i) The current Table 1 is unclear and confusing. In Table 1, it would be more informative if the comparative proportion of individuals is reported by row and not by column. For instance, how many proportions of males had/did not undergo preventive scaling, etc. Also, Table 1 needs to be re-organized into categorical variables, which is reported as N (%) and continuous variables, reported as Mean (SD).

ii) In Tables 1 and 2, remove the term “baseline” as this is a cross-sectional data with no baseline and follow-ups.

iii) In the Discussion, the authors need to discuss the potential role of dental visits for more complex procedures (such as surgical/non-surgical periodontal therapy) in those with metabolic syndrome and/or unhealthy lifestyle, which would confound the absence of codes for preventive scaling in this group of people.

iv) Also, potential explanations for the associations observed in the study needs further corroboration with evidence from literature.

Reviewer #3: This is an interesting manuscript on the association between metabolic syndrome, unhealthy lifestyle habits and preventive dental scaling using a huge dataset from Korea. As results, authors showed that these characteristics are interconnected and clustered in the sample, and that dose-response relationships with dental scaling for preventive purposes occur.

Introduction:

*There are some sentences that called my attention:

"Calculus, which is not removed by routine oral hygiene practices such as brushing, is the CAUSE of periodontitis along with the bacterial plaque that covers it."

"Periodontitis bacteria in dental calculus and bacterial plaque locally DESTROY not only the soft tissues around the teeth but also the alveolar bone, resulting in tooth loss."

"Periodontal pathogens, which CAUSE periodontal disease..."

In my opinion, these sentences should be accurately reviewed. Current knowledge shows that periodontitis causation are not so simply as stated.

*State study's hypotheses.

Methods:

*It would be interesting to see how variables were hierarchically organized. Consider to use a flowchart for that.

Results:

*I am not convinced on the way how authors reported their results. In table 1, for instance, authors showed all independent variables assessed, and described these as "baseline characteristics". In table 2, they also called "baseline characteristics" but selected some to report. Additionally, tables 3 and 4 showed some data also previously presented in table 2 (as the effect measures are the same). For instance, high blood pressure/hypertension, drinking, smoking and regular exercise.

My point is that these tables can be simplified and data could be accommodated in two tables (one for unhealthy lifestyle and another for metabolic syndrome).

*As authors described dose-response relationships, I would suggest to remove such findings from tables and present the odds ratios graphically.

Discussion:

*Authors centered the discussion (and introduction as well) on the microbiological path that links exposures and periodontitis. What about the inflammatory pathway?

*Indeed, what about behavioral pathways? Authors described that metabolic syndrome components, unhealthy lifestyle habits and lack of preventive oral care are clustered in the same individuals. What health professionals should do in this respect?

*Language revision would be useful to avoid some redundancies such as "treatment to treat".

6. PLOS authors have the option to publish the peer review history of their article (what does this mean?). If published, this will include your full peer review and any attached files.

Reviewer #1: No

Reviewer #2: No

Reviewer #3: **Yes: **Leandro Machado Oliveira

---

## [Author Response · Author response to Decision Letter 0]

15 Aug 2023

I appreciate your kind advice about this manuscript. I did my best to correct the flaws. The corrected items were marked as yellow color in the manuscript. Please let me know if there are any more revisions.

---

## [Decision Letter · Decision Letter 1]

13 Sep 2023

PONE-D-23-15887R1Association between dental scaling and metabolic syndrome and lifestylePLOS ONE

Dear Dr. Yun,

Thank you for submitting your manuscript to PLOS ONE. After careful consideration, we feel that it has merit but does not fully meet PLOS ONE’s publication criteria as it currently stands. Therefore, we invite you to submit a revised version of the manuscript that addresses the points raised during the review process.

Dear authors,

Thank you for sending your revised manuscript.

While the referees have noticed an improvement in your manuscript, inconsistencies remain to be further addressed. The definition of periodontitis merits attention and should be revised according to the current understanding of the disease.

In addition, please clarify some methodological aspects, such as the nature of the periodontal treatment variable (e.g., binary or discrete).

Thank you.

We look forward to receiving your revised manuscript.

Kind regards,

Gustavo G. Nascimento, PhD

Academic Editor

PLOS ONE

Journal Requirements:

Reviewers' comments:

Reviewer's Responses to Questions

**Comments to the Author**

1. If the authors have adequately addressed your comments raised in a previous round of review and you feel that this manuscript is now acceptable for publication, you may indicate that here to bypass the “Comments to the Author” section, enter your conflict of interest statement in the “Confidential to Editor” section, and submit your "Accept" recommendation.

Reviewer #1: (No Response)

Reviewer #2: All comments have been addressed

Reviewer #3: (No Response)

2. Is the manuscript technically sound, and do the data support the conclusions?

Reviewer #1: Yes

Reviewer #2: Yes

Reviewer #3: Yes

3. Has the statistical analysis been performed appropriately and rigorously? 

Reviewer #1: Yes

Reviewer #2: Yes

Reviewer #3: Yes

4. Have the authors made all data underlying the findings in their manuscript fully available?

Reviewer #1: Yes

Reviewer #2: Yes

Reviewer #3: Yes

5. Is the manuscript presented in an intelligible fashion and written in standard English?

Reviewer #1: Yes

Reviewer #2: Yes

Reviewer #3: No

6. Review Comments to the Author

Reviewer #1: It must be acknowledged that the authors have made progress in the construction of the text and have greatly improved the way it is presented. Nevertheless, I suggest clarifying some of the study's important points.

Abstract/Results

The sentence: "Multiple logistic regression analysis showed that after adjustment for age, sex, income, body mass index (BMI), smoking, drinking and regular exercise, the Odds ratios (OR) and 95% confidence intervals (CI) of" is part of the methods and not the results.

Introduction

The authors approached a quite obvious pathway in which periodontal bacteria would cause cardiometabolic diseases. But right after that, the authors say sentences like “It has been reported that people with metabolic syndrome are twice as likely to develop periodontitis than those without it” demonstrating that perhaps periodontitis is caused by these diseases. It seems more plausible to believe that both periodontitis and cardiometabolic diseases have common causes such as smoking, alcohol, diet, etc. As there is no consensus on this, this should be expressed in the text. In the last two hypotheses, the low demand for dental scaling would be a clear demonstration that individuals have an unhealthy lifestyle, both in terms of general health and oral health.

Methods

The outcome variable preventive dental scaling needs to be better defined. Firstly, it needs to be stated in the text that this is the outcome studied, for example, "The outcome variable in this study was the preventive dental scaling...". Subsequently, the authors do not make it clear how this very important variable in the study was measured. Apparently, it was considered as a binary (y/n) variable, but this is not stated in the text. Furthermore, I have some doubts: Was it considered as “at least one visit”? Was it considered as the total number of visits (continuous)?

Were participants who were referred directly for therapeutic dental scaling excluded from the analysis? This may seem obvious, but it is important to clarify in the text, since the participant may not have carried out the preventive phase because their periodontal health had already deteriorated and therefore the preventive approach would no longer make sense in this situation.

Understanding that the frequency of visits for periodontal treatment is strongly linked to a healthy lifestyle, performing models adjusted for the number of medical visits (or another healthy lifestyle indicators) is necessary to help control confounding bias.

Discussion

The sentence: “Periodontitis is a chronic inflammatory infectious disease” does not make sense based on the WHO division of diseases, i.e. chronic (chronic inflammatory) or infectious diseases. From the relationship between periodontitis and other chronic diseases, it seems to be more of a chronic disease than an infectious disease, which is confirmed in the first 6 lines of the third paragraph of the discussion. Bear in mind that the relationship between periodontitis and other diseases is not clear from the literature.

P17: “The present study also investigated how many times the people with bad lifestyle habits were dental scaling.” The authors do not mention in the methodology that the outcome was considered to be the number of times the participant underwent (continuous) treatment. If dental scaling was considered as being binary, this sentence should be rewritten, as what was assessed was the probability/odds of having or not having undergone dental scaling.

Reviewer #2: (No Response)

Reviewer #3: The authors are commended for their thoughtful review of the manuscript. Few issues remain to be addressed in the manuscript:

Introduction:

*There are STILL some sentences that called my attention:

"Calculus, which is not removed by routine oral hygiene practices such as brushing, is the CAUSE of periodontitis along with the bacterial plaque that covers it." - you may consider dental plaque (dysbiosis) as a causal component of periodontitis, however, to the best of my knowledge, there is no evidence to support such a role for calculus;

"Periodontitis bacteria in dental calculus and bacterial plaque locally DESTROY not only the soft tissues around the teeth but also the alveolar bone, resulting in tooth loss." - periodontopathogens are responsible for a small part of tissue destruction. The major contributor is host immune response;

"Periodontal pathogens, which CAUSE periodontal disease..." - current knowledge shows that periodontitis causation are not so simply as stated. Please, follow current definitions of periodontitis etiology.

Methods:

*Authors described their modeling strategy. However, there is no clear explanation on why such arrangement was done. For instance, why "unhealthy lifestyle" variables were combined with the socioeconomic ones for metabolic syndrome, and another block (exclusive for comorbidities) was used when unhealthy lifestyle was the outcome? Additionally, BMI is not a socioeconomic variable.

*Was there any criteria for variable retention? Or all variables were included?

Discussion:

*Language revision would still be useful.

"(...) preventive treatment to prevent (...)"

"(...) people with bad lifestyle habits were dental scaling."

7. PLOS authors have the option to publish the peer review history of their article (what does this mean?). If published, this will include your full peer review and any attached files.

Reviewer #1: No

Reviewer #2: No

Reviewer #3: **Yes: **Leandro Machado Oliveira

---

## [Author Response · Author response to Decision Letter 1]

24 Oct 2023

I appreciate your kind advice about this manuscript. I did my best to correct the flaws. The corrected items were marked as yellow color in the manuscript.

---

## [Editor Report · Decision Letter 2]

26 Oct 2023

PONE-D-23-15887R2Association between dental scaling and metabolic syndrome and lifestylePLOS ONE

Dear Dr. Yun,

Thank you for submitting your manuscript to PLOS ONE. After careful consideration, we feel that it has merit but does not fully meet PLOS ONE’s publication criteria as it currently stands. Therefore, we invite you to submit a revised version of the manuscript that addresses the points raised during the review process. Dear authors, Thank you for submitting your revised manuscript. I read with interest the corrections you have made. While the methodology became clearer in this version, the Introduction section, more specifically, the concepts of periodontitis causation and its relationship with systemic diseases, should be carefully revised. While a dysbiotic biofilm has been associated with periodontitis, it has been known that plaque deposits account for approx. 20% of periodontitis causation, while the remaining 80% relate to the host's ability to deal with the inflammatory process. You can find more details elsewhere (1). On a similar note, the authors assume periodontitis as a cause for several systemic diseases, such as endothelial dysfunction, myocardial infarction, and diabetes, among others. While studies have shown associations between these conditions, causation still neeeds to be established (2). In addition, referee #3 mentioned that a potential link between periodontitis and systemic diseases relies on their shared risk factors, such as smoking and diet, among others. This has not been included in the manuscript, and, given the aim of this study - to investigate lifestyle factors in addition to MetS - this should be included and discussed (3).  Once you revise your theoretical background accordingly, a final editorial decision can be made.   Thank you for taking an extra mile. Kind regards,Gustavo.  References: 1. Bartold PM, Van Dyke TE. An appraisal of the role of specific bacteria in the initial pathogenesis of periodontitis. *J Clin Periodontol*. 2019;46(1):6-11. doi:10.1111/jcpe.130462. Raittio E, Farmer J. Methodological Gaps in Studying the Oral-Systemic Disease Connection. *J Dent Res*. 2021;100(5):445-447. doi:10.1177/00220345209829723. Frandsen Lau E, Peterson DE, Leite FRM, et al. Embracing multi-causation of periodontitis: Why aren't we there yet?. *Oral Dis*. 2022;28(4):1015-1021. doi:10.1111/odi.14107

We look forward to receiving your revised manuscript.

Kind regards,

Gustavo G. Nascimento, PhD

Academic Editor

PLOS ONE
---

## [Author Response · Author response to Decision Letter 2]

28 Nov 2023

I appreciate your kind review and comments. Following your advice, we've revised the Introduction section to reflect the recent articles.

---

## [Editor Report · Decision Letter 3]

9 Jan 2024

Association between dental scaling and metabolic syndrome and lifestyle

PONE-D-23-15887R3

Dear Dr. Yun,

We’re pleased to inform you that your manuscript has been judged scientifically suitable for publication and will be formally accepted for publication once it meets all outstanding technical requirements.

Kind regards,

Gustavo G. Nascimento, PhD

Academic Editor

PLOS ONE
---

## [Editor Report · Acceptance letter]

25 Mar 2024

PONE-D-23-15887R3 

PLOS ONE

Dear Dr. Yun, 

I'm pleased to inform you that your manuscript has been deemed suitable for publication in PLOS ONE. Congratulations! Your manuscript is now being handed over to our production team.

Kind regards, 

on behalf of

Dr. Gustavo G. Nascimento 

Academic Editor

PLOS ONE